# Double Pre-Bending of an Intramedullary Nail Is the Minimal Invasive Osteosynthesis Solution for Dia-Metaphyseal Fractures of the Radius in Children: Technical Note and Case Series

**DOI:** 10.3390/children9040579

**Published:** 2022-04-18

**Authors:** Carsten Krohn

**Affiliations:** Deputy Chief Department of Paediatric Surgery, München Klinik Schwabing, Kölner Platz 1, D-80804 München, Germany; carsten.krohn@muenchen-klinik.de

**Keywords:** dia-metaphyseal, fracture, radius, child, adolescent, intramedullary nail

## Abstract

Whereas in paediatric traumatology for diaphyseal fractures of the radius (intramedullary nail), as well as metaphyseal fractures (K-wire), minimal invasive methods for osteosynthesis were established as gold standard, the ideal osteosynthesis of fractures of the dia-metaphyseal area remains controversial. In this article, the author describes his own minimal invasive technique, using an intramedullary nail that must be pre-bent twice to achieve a stable reduction, with three-point support. The material used for this new surgical technique is an ordinary intramedullary nail. If not available, the operation can also be performed with a sufficiently lengthy K-wire. The intramedullary nail needs to be pre-bent twice, which follows a standardised procedure. A small case series is included to visualise the scope of this minimal-invasive method.

## 1. Introduction

The treatment of dia-metaphyseal fractures has been discussed controversially over the last decades [1]. Conservative treatment, including reduction and retention with the use of proper cast immobilisation, should be the treatment of first choice in stable fractures [2]. However, secondary displacement is common and, therefore, preferably minimal-invasive osteosynthesis should be used in instable fractures [3,4,5]. Whilst in diaphyseal fractures, a K-wire osteosynthesis is the most favoured option, and in diaphyseal fractures, intramedullary nailing seems to be the treatment of choice worldwide, searching for an ideal minimal-invasive technique in dia-metaphyseal fractures is an ongoing procedure. This “region of interest”, the dia-metaphyseal transition zone, is defined as the square over the “physis of distal radius and ulna” minus the square of “physis of the distal radius alone” [1]. Here, plate osteosynthesis provides a well-established but invasive technique, which is a standard procedure in adult trauma surgery but should be avoided, if possible, in growing bones in childhood and adolescence, and is also not free of complications [6,7]. External fixation devices should be in everyone’s repertoire, e.g., for open fractures; however, the author feels that, due to the need for four incisions and the inconvenience caused by the external fixation device in children, this does not fulfil the criteria of a minimal-invasive technique [6]. K-Wires, being drilled into the bone retrograde from the styloid process of the distal radius, may not provide enough stability and require high surgical skills in this particular fracture. Insufficient K-wire osteosynthesis is prone to complications [6] (see Figure 1). However, if suitable, this is a minimal-invasive technique, which can produce perfect results [8,9].

Intramedullary nails, however, if not pre-bent, usually lead to an unacceptable radial displacement of the distal fragment, which makes the situation sometimes worse than it was before (Figure 1). Pre-bending techniques have been described with a single kink, which is placed distal to the fracture, but these remain controversial due to lack of stability [3,10,11]. The same applies to unconventional techniques, such as locked wires [12].

In 2011, Lieber and Sommerfeldt stated that intramedullary nailing is not ideal for the osteosynthesis of dia-metaphyseal fractures in childhood and adolescence due to biomechanical problems [6].

This article presents a new minimal-invasive method of intramedullary nailing, bending an intramedullary nail twice before insertion. This influences the biomechanical properties of the intramedullary nail and achieves a stable and anatomically correct osteosynthesis of dia-metaphyseal fractures in childhood and adolescence.

## 2. Surgical Technique

The material that is used for this new surgical technique is an ordinary intramedullary nail—if not available, the operation can be performed with a strong and sufficiently long K-wire (blunt end being bent like the skid of an ESIN and inserted first) as well. This intramedullary nail needs to be pre-bent, which follows a standardised procedure that will be described in this article. Due to the fact that steel, when bent, retains the kink more reliably than titanium, a steel nail is preferred to achieve a stable osteosynthesis [13,14].

The idea that this new surgical technique is based on is that, if a normal intramedullary nail being brought into the radius from the radial side of the distal radius, displaces the distal fragment radially, the solution would be to insert the nail from the ulnar side, which, however, is not possible, because this area cannot be accessed due to the close anatomical neighborship of the distal ulna. This makes it necessary to simulate an ulnar access. Therefore, the proximal (first) kink of the nail, which at the end of the procedure is placed proximal to the fracture, will push the radial corticalis of the proximal fragment radially, whereas the second kink (which can also be performed as a bend, see Figure 1) pushes the distal fragment into an anatomical position and makes it possible for the nail to be brought into the distal radius using the established approach. If the entry into the bone is performed exactly with the diameter of the nail, stable osteosynthesis with three-point support is achieved (Figure 1).

The surgical technique (see Table 1) must be performed in a standardised way:

First, the surgeon must determine if the fracture is really suitable for this technique: all fractures of the dia-metaphyseal junction are suitable if not open and contaminated. We have not seen this type of fracture in pre-existing bone diseases, such as osteogenesis imperfecta, and cannot give advice as to whether this technique is suitable in pathologic fractures. Dia-metaphyseal fractures combined with a fracture of the ulna are absolutely suitable. In this case the author prefers to stabilize the ulnar fracture first with an intramedullary nail. Here, the diameter should be chosen to be as large as possible. The method of stabilisation of an ulnar fracture has been described in many textbooks; therefore, only the technique that should be used for repair of the radius is described here.

The patient is placed in supine position and the fractured arm is prepped in a 90° abduction on a hand table. The fluoroscopy is placed cranially and can be moved parallel to the patient.

After choosing the right diameter (which should not be as big as possible: about half the diameter of the intra-cortical space is ideal), the distance between skid and the proximal kink must be established. To avoid x-ray exposure to the surgeon’s hand, the nail is grasped with forceps and is placed on the patient’s arm, skid facing upwards. Under fluoroscopy, the position of the first kink is determined about 1 cm proximal to the fracture (skid must be distal to the growth plate of the proximal radius) and is marked by bending the nail slightly (the final bend of about 45° should be performed later: if it is performed now, the nail will not fit into the handpiece, see Figure 2). Now, a skin incision is performed, lateral of the distal radial growth plate, and the radial metaphysis is perforated, just proximal to the growth plate, as in ‘normal’ intramedullary nailing. The nail is introduced into the bone until the kink appears between the bone and handpiece: now the kink is completed to at least 45°. This is the important moment; the second kink must be performed, about 2–3 cm distal to the first kink (the distance between the kinks equals about twice the diameter of the radius in the region of the fracture), in the opposite direction to the first kink, again at least 45° (Figure 2). The nail can be bent to modify the distal kink (Figure 1); however, the aim is that this kink finally gets contact to the ulnar-sided cortical of the radius and leads to a tension-free curve to the area of insertion.

Now, the double-kinked nail must be advanced into the bone, which might be a little tricky and require some surgical skills. It may be necessary to both push and rotate the nail whilst bringing it up to the radial neck. Finally, when the skid is placed in the radial neck, the proximal kink should be proximal to the fracture and the second kink distal to it. Rotating the nail this way, so that the proximal kink touches the radial-sided cortical and the distal kink the ulnar-sided cortical of the radius, will achieve reduction and retention of the fracture. If no anatomically correct position is achieved, the kinking of the nail might not be sufficient. Furthermore, there are still spontaneous correction abilities of the growing bone to be taken into account. In some cases, the proximal kink stretches out during insertion and cannot be detected on the final X-ray; in case of a sufficient reduction, this should be accepted and does not cause any worries. To finalise the operation, the nail should be shortened, the wound closed, and a final X-ray should be performed to document the proper reduction. A two-week forearm immobilisation should be performed for pain relief and to minimise the risk of nail flip. Even though we have never seen a nail flip (with loss of reduction and secondary displacement), this risk should be taken into account.

## 3. Case Series

As a kaleidoscope of possibilities as to which fractures can be treated with this minimal-invasive technique, some results of my procedure are shown. There were no cases requiring open reduction or a need for a change in procedure. All patients that returned to our hospital for removal of the nail(s) had full range of movement and, thus, required no follow-up examinations.

The youngest patient was a two-year old boy, who presented with an unacceptable secondary displacement two weeks after trauma and, thus, this fracture became a rare indication for osteosynthesis in this age group (Figure 2). The eldest was 17 years of age and did not show epiphyseal plates anymore (Figure 3). This patient, however, suffered a new trauma 3 weeks later and underwent a redo-procedure in another hospital and, thus, was lost to follow-up. We even treated an open forearm fracture (on the ulnar side) with this technique (Figure 4), leaving the fracture of the ulna to spontaneous healing and correction. This was because the risk of osteomyelitis due to intramedullary nailing of the open ulnar fracture to achieve a “nice X-ray” was estimated to be much higher than the risk of a remaining misalignment. Even a redo-procedure of a fracture that we had seen for the first time 4 weeks after an unacceptable K-wire osteosynthesis was successful, this time using a modification of the technique with a bend, rather than a kink distal to the fracture. Please note that the pre-bent nail pushed the 4-week-old fracture into a correct position without open reduction (Figure 4). Last but not least, curiously, we saw one patient who suffered identical fractures in both arms, so we could perform the procedure twice in one operation (Figure 5).

**Scheme 1 children-09-00579-sch001:**
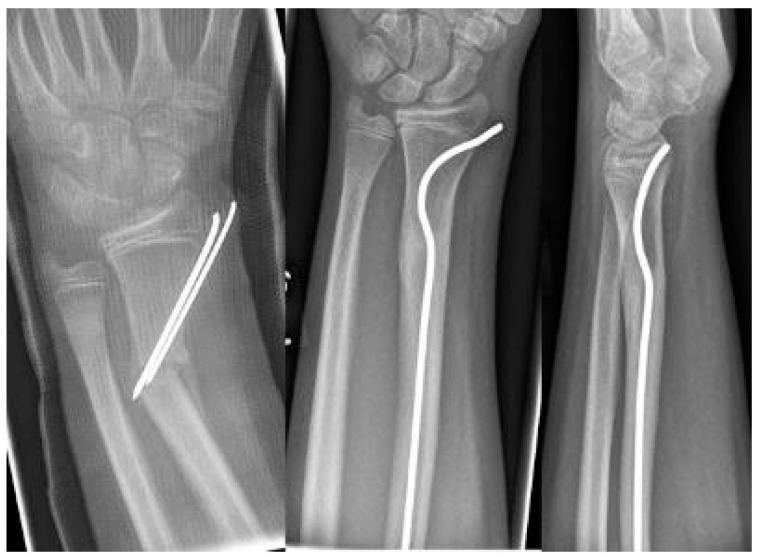
Closed reduction 4 weeks after insufficient k-wire osteosynthesis with a bend instead of the distal kink as a variant of this procedure.

**Scheme 2 children-09-00579-sch002:**
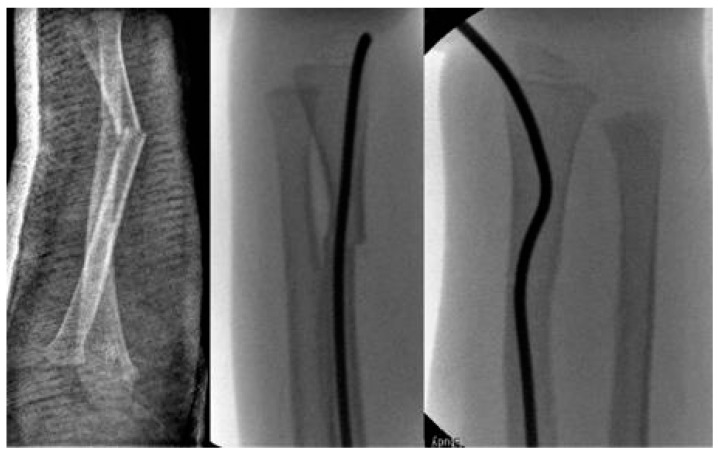
2-year-old boy with secondary displacement.

**Scheme 3 children-09-00579-sch003:**
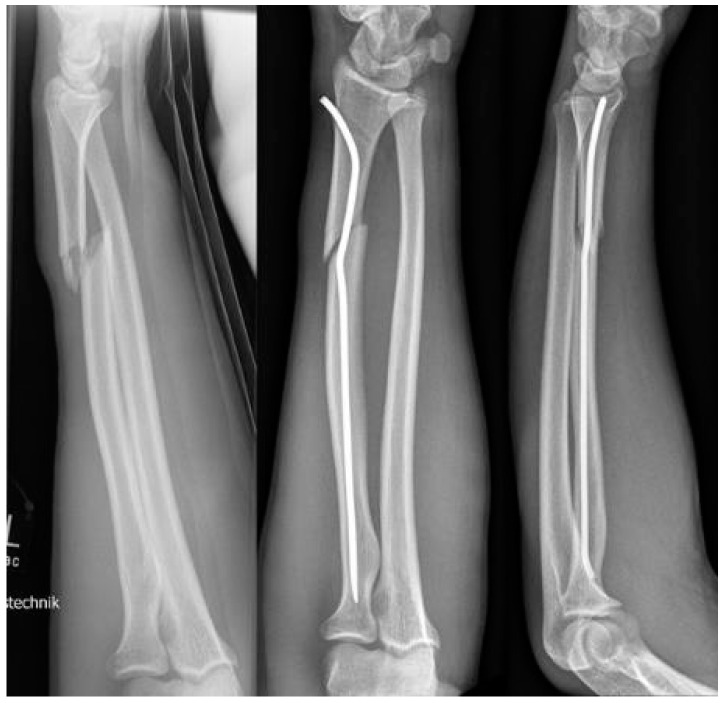
17-year-old boy.

**Scheme 4 children-09-00579-sch004:**
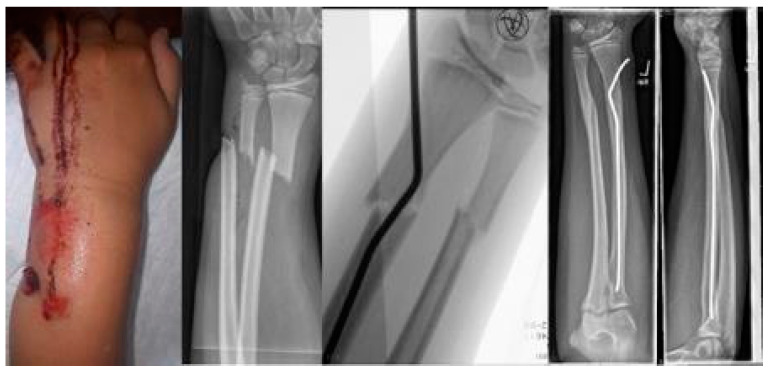
Open forearm fracture leaving the fracture of the ulna to spontaneous correction.

**Scheme 5 children-09-00579-sch005:**
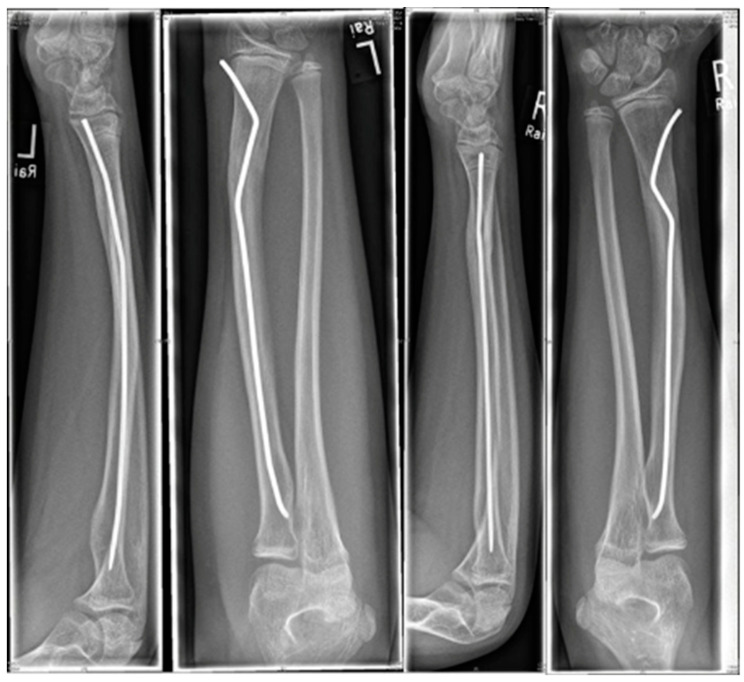
Patient with identical fractures in both arms.

## 4. Discussion

The delivery of good medical care is to do as much of nothing as possible [15]; thus, minimal-invasive therapy should be the approach of all surgeons for the treatment of paediatric patients. Therefore, closed reduction and immobilisation in a proper cast should be the therapy of choice in any paediatric fracture, whenever appropriate [2]. If stability cannot be achieved or if secondary displacement occurs, osteosynthesis is necessary and should be as minimally invasive as possible [1]. A well-placed K-wire, which does not need to perforate the cortical bone proximal to the fracture but can be placed intramedullary in the proximal fragment, is a good and minimal-invasive technique that can solve the problem [8]. However, the proper placement of this K-wire is sometimes impossible. Therefore, we developed a method using an ESIN. If used without pre-bending, an anatomical reduction can never be achieved. The double pre-bending technique has been developed in our hospital for more than ten years and is the standard technique for this specific fracture. If the entry into the bone is performed exactly with the diameter of the nail, stable osteosynthesis with three-point support is achieved, which makes it superior to other pre-bending and non-pre-bending intramedullary techniques. It can even be used in redo-procedures after failed conservative treatment or K-wire osteosynthesis. If minimal displacement remained after inserting the nail, which is known to heal with spontaneous correction of the growing bone [6], it was accepted. This is why closed epiphyseal plates are a limit for this procedure; those patients should be treated according to the rules of adult traumatology. There were no conversions to another procedure, e.g., plates or external fixation. However, if a good reduction cannot be achieved, a change of procedure using a plate must be taken into consideration, which is a well-established, but not a minimal-invasive, technique. We have not yet treated any patient with congenital bone disease (e.g., osteogenesis imperfecta or juvenile cysts), so no recommendations can be given for this group of patients. By no means can this surgical technique be successfully performed in open and contaminated wounds that require treatment with an external fixation device. 

The author is convinced that the double pre-bent ESIN is the only truly minimal-invasive and sufficiently stable alternative for osteosynthesis of closed dia-metaphyseal radial fractures to a well-placed K-Wire.

This article is a technical note, including a case series, and is not a scientific study. The aim of this article is to spread this technique among surgeons. The number of patients with this fracture entity is very small, even in specialised hospitals, so further (multicentre) studies are necessary to evaluate the value of this technique.

## Data Availability

Not applicable.

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
