# Peer review of "Double Pre-Bending of an Intramedullary Nail Is the Minimal Invasive Osteosynthesis Solution for Dia-Metaphyseal Fractures of the Radius in Children: Technical Note and Case Series"

_children, 2022, doi:10.3390/children9040579_

Round 1

Reviewer 1 Report

  1. The Technique is novel and minimally invasive.
  2. Novel procedure seems to be more difficult than the previous subscribed.  Technical difficulties, two bending, one proximal and one distal of the fracture line. If the kink is not in the correct place it can cause dislocation or instability. 
  3. The steps are clear but not easy and you do not standardise how to bend exactly, it is always individual. Differ for transverse and spiral fracture or in both forms it could cause the same stability? 
  4. In my opinion this technique is not easier or lurnable than the well known described and widely used prebending intramedullary nailing written by Teddy Slongo.  I can not see any advantages of this new technique. 
  5. How many complications did you have during the operation, did you have the case when you had to change the surgical method? HAve you noticed the opposite cortical fracture? Have you had any experience of the pre bending straight during the time of insertion? 
  6. The fixation (two kinks) is probably stable in one plane, sagittal or frontal as all intramedullary techniques have the same problem. 

Author Response

Thank you for revising my article: all three reviewers obviously appreciate my idea and new surgical approach as novel and suitable for everyday use in pediatric traumatology. All suggestions about what I could and should change are well understandable to me. Being a “normal” pediatric surgeon, I feel supported by all three reviewers even though I am not a part of the specific scientific community and as you might have noticed, my experience in publishing is low. All reviewers are supportive to my efforts to publish my idea, which makes me feel grateful.

Before I answer to each reviewer I want to point out, that my article was never intended to be a scientific study. This article is the summary of the development of a new method, which, to be honest, was only recently realized by myself as a method, that can (and I hope it will) change the approach to dia-metaphyseal fractures of the radius in children and adolescents. This is, why this article is a technical note. In addition, I do present some of the cases I collected during the last years. I could have tried to publish two articles: A technical note and a case series according to this technical note. However: I feel, that this belongs together and thus I would appreciate if you would accept my case series being imbedded into the technical note.

The next step will and should be to evaluate the outcome in a multicenter trial and collect patient data and not only describe a new method.

  1. Thank you!
  2. True: If the kink is in the wrong space, the nail must be discarded. This is, why the place for the kink hast to be established with fluoroscopy (see Figure 2 and Table 1)
  3. Spiral fractures in this area are very rare. I cannot remember even one. In case of a spiral fracture with a long spiral – if that occurs in the metaphyseal region – the fracture might need a plate or an external fixation. As mentioned: I cannot remember a spiral fracture in this region.
  4. Theddy Slongo is one of my heroes concerning paediatric traumatology. However: I feel that “my method”, to double kink an intramedullary nail, results in higher stability. I would be happy if you’d try… Finally, a further study should try to determine the difference in outcome of the different methods…
  5. This question is answered in the article: “There were no conversions to another procedure e.g. plates or external fixation.”
  6. This is one of the unsolved problems. As mentioned in the article, I do fear the “flip” of the nail even though I never saw one. Therefore I suggest a cast immobilisation for 2 weeks.

Reviewer 2 Report

the technique is presented very well, but there schould be numbers treated in this hospital with this method. They write it is the "standart procedure", but they should tell us about the outcome of patients treated ( Numer Age, range, lenghth of imobilisation, length of material, and outcome after operation.

Author Response

Thank you for revising my article: all three reviewers obviously appreciate my idea and new surgical approach as novel and suitable for everyday use in pediatric traumatology. All suggestions about what I could and should change are well understandable to me. Being a “normal” pediatric surgeon, I feel supported by all three reviewers even though I am not a part of the specific scientific community and as you might have noticed, my experience in publishing is low. All reviewers are supportive to my efforts to publish my idea, which makes me feel grateful.

Before I answer to each reviewer I want to point out, that my article was never intended to be a scientific study. This article is the summary of the development of a new method, which, to be honest, was only recently realized by myself as a method, that can (and I hope it will) change the approach to dia-metaphyseal fractures of the radius in children and adolescents. This is, why this article is a technical note. In addition, I do present some of the cases I collected during the last years. I could have tried to publish two articles: A technical note and a case series according to this technical note. However: I feel, that this belongs together and thus I would appreciate if you would accept my case series being imbedded into the technical note.

The next step will and should be to evaluate the outcome in a multicenter trial and collect patient data and not only describe a new method.

Thank you for appreciating my method as a good presentation of the technique. As it is a technical note and case series, it lacks of patient data, because they weren’t collected in a proper, scientific way (as mentioned above, the fact, that this really is a new idea and can give an input to other surgeons, was noticed only recently. A publication was not intended, when I started to develop this method. If I only had realised that earlier…).

So this article is intended to be the birth of the idea to a broad community of surgeons. This is, why it is “only” a technical note (and case series) and not an “original article”. I absolutely agree you are right: The next step must be the collection of patient data: prospective – and if possible multicenter!

Reviewer 3 Report

This manuscript by Krohn C reports a technical note describing an modified ESIN-osteosynthesis for pediatric dia-metaphyseal forearm/radius fractures. This technique is the result of previous attempts to establish a stable osteosynthesis for unstable fractures in the diametaphyseal region, which has fewer or even no disadvantages in terms of invasiveness, technical difficulty, complication rate, and others compared to already described techniques.

The findings of this article are very important to all practitioners dealing with pediatric fractures and it offers a solution for a problem, for which research has been performed for many years. However, the manuscript in its current form does not meet the requirements for a "Technical Note". Usually there are no "Results" in a Technical Note, but only the subheadings "Introduction", "Method" or "surgical technique", and "Discussion". At the moment, the formatting does not correspond to a scientific publication. For example, the abstract already reports "all patients… had a full range of motion…", whereas no patient collective is described in the entire manuscript (number of patients, mean age, follow-up data e.g.). The "Results" consist of illustrating various (incomplete) cases, there are no standardised follow-up examinations, some patients have no follow-up examinations at all.

The author must be given the opportunity to creat and illustrate a "Technical Note". According to the MDPI website, this is possible, but the reviewer recognises that a technical report has not yet appeared in "Children". At the very least, the author must be allowed to use the above-mentioned (sub)headings. Additionally, these minor points should be revised:

„Title“

The title is incorrect as it mixes the entities "dia-metaphyseal" and "distal" fractures. Correct would be: Double pre-bending of an intramedullary nail is a minimal invasive osteosynthesis solution for pediatric dia-metaphyseal radius fractures.

“Abstract”

The report of patient data should be removed from this section as it is a "technical note".

„Introduction“

At least the definition of "dia-metaphysis" should be mentioned or introduced with a label in an X-ray.

The claim that an external fixator is "invasive" (line 33) is false and the opposite is true. But the procedure has certain disadvantages for the patient when it comes to the treatment of a dia-metaphyseal forearm fracture.

Figures 1, 2 and 3 play no role in the description of the double pre-bended ESIN osteosynthesis technique and should be removed. The description of the disadvantages of these methods in the "Introduction" is sufficient.

„Method“ or „surgical technique“

The detailed description of the surgical technique is sufficient; Figure 6 is redundant. Figure 7 is irrelevant in term of the surgical technique. The author already describes in the manuscript that this is a technical variation of the established ESIN osteosynthesis. A standard patient and fluoroscopy positioning has already been described in many textbooks and articles.

Figures 5, 8 and 9 are sufficient to illustrate the surgical technique. For this purpose, another 1-2 cases should be illustrated. The author should ensure that these cases are dia-metaphyseal fractures. Figure 15 is by definition a distal shaft fracture.

„Discussion“

A statement should be made on the biomechanical advantages of the presented technique compared to classical ESIN osteosynthesis. Furthermore, the advantages compared to alternative techniques should be mentioned. The range of indications should also be discussed. This means, for example, case/figure 14 (change of method), 11 (secondary dislocation) and 14 (closed growth plates).

 „Conclusion“

No more references are mentioned in the conclusion-section.

In summary, this paper is highly recommended to be published, because this surgical technique sounds very effective for the problematic dia-metaphyseal forearm region. Therefore, the reviewer would very much appreciate to give the author the opportunity to revise the manuscript as a “technical note”.

Author Response

Thank you for revising my article: all three reviewers obviously appreciate my idea and new surgical approach as novel and suitable for everyday use in pediatric traumatology. All suggestions about what I could and should change are well understandable to me. Being a “normal” pediatric surgeon, I feel supported by all three reviewers even though I am not a part of the specific scientific community and as you might have noticed, my experience in publishing is low. All reviewers are supportive to my efforts to publish my idea, which makes me feel grateful.

Before I answer to each reviewer I want to point out, that my article was never intended to be a scientific study. This article is the summary of the development of a new method, which, to be honest, was only recently realized by myself as a method, that can (and I hope it will) change the approach to dia-metaphyseal fractures of the radius in children and adolescents. This is, why this article is a technical note. In addition, I do present some of the cases I collected during the last years. I could have tried to publish two articles: A technical note and a case series according to this technical note. However: I feel, that this belongs together and thus I would appreciate if you would accept my case series being imbedded into the technical note.

The next step will and should be to evaluate the outcome in a multicenter trial and collect patient data and not only describe a new method.

Thank you for your statement The findings of this article are very important to all practitioners dealing with paediatric fractures and it offers a solution for a problem, for which research has been performed for many years.”

According to your suggestions I changed:

  1. Subheadings are changed to Introduction / Surgical technique / Discussion according to your suggestions. The Examples I want to present are listed in a subheading “Case series”. There are no “Results” or “Conclusions” to be found anymore. I hope, this meets the requirements to a technical note and case series.
  2. The title has been changed according to your suggestions
  3. Patient “data” has been removed according to your suggestions and changed into “Case series” – I hope you can accept this as illustration of the technical note
  4. The definition of “dia-metaphyseal region” has been included into the introduction according to your suggestions
  5. We disagree concerning the question if an external fixation device is minimal invasive or not. This is why I included the sentence: “External fixation devices should be in everyone’s repertoire, e.g. for open fractures; however, the author feels that due to the need of 4 incisions and the inconvenience caused by the external fixation device in children this does not fulfill the criteria of a minimal invasive technique”
  6. Figures 1-3 have been removed according to your suggestions
  7. Figure 7 has been removed according to your suggestions
  8. Figure 15 has been removed according to your suggestions
  9. Discussion has been modified according to your suggestions
  10. Conclusion has been included into discussion, because you did not want it as subheading.

In summary your review was very helpful to help me to convert my article into a technical note and I hope that I met your expectations!

Round 2

Reviewer 1 Report

I accepted the Author's letter and I understood that it was a technical description. Sometimes it is difficult to publish like this study but other two reviewers suggested after the major revision. I appreciate the decision. 

After correction I accept.

If is is possible I would be grateful to see this method during the live surgery. 

Best

Reviewer 2 Report

thx for editing the paper

Reviewer 3 Report

This manuscript and has been revised according to the given remarks and it has gained relevance now to be considered for publication in Children.